# Design and Experiment of Black Soldier Fly Frass Mixture Separation through a Cylinder Sieve with Different Rotation Speeds

**Caiwang Peng [1], Ting Zhou [1], Shisheng Song [1], Songlin Sun [1], Yulong Yin [2,\*] and Daojun Xu [3,\*]**

1   College of Mechanical and Electrical Engineering, Hunan Agricultural University, Changsha 410128, China
2   College of Animal Science and Technology, Hunan Agricultural University, Changsha 410128, China
3   College of Veterinary Medicine, Hunan Agricultural University, Changsha 410128, China
\*   Correspondence: yinyulong@isa.ac.cn (Y.Y.); xudaojun29@163.com (D.X.)

**Featured Application: In this study, a differential separation roller screen was developed, and the nail teeth and the screen were rotated coaxially with reverse directions. The mixture of black soldier fly sand with a certain moisture content was more effectively separated, and the artificial separation intensity for this mixture was reduced, thus facilitating the classification and utilization of black soldier fly larvae and organic fertilizer of insect sand.**

**Abstract:** A differential separation trommel screener was developed to solve the problems of the impurity content in insects and the high rate of insect impurities in the separation of black soldier fly (BSF) sand mixture. Moreover, the mechanical and physical properties of the BSF sand and its bonding contact model were examined. With the rotational speed of the trommel and the spikes and the inclination of the trommel as the experimental factors, their motion characteristics were analyzed and their value ranges were determined. In addition, the impurity content in the insects and the rate of insect impurities were selected as the test indicators. The Box–Behnken test was performed, the response surface regression model was built, and the parameters were optimized. The results indicated that the respective test factors, the impurity content and the insect rate, followed the following order of significance: the trommel rotation speed, spike teeth rotation speed, and trommel screener inclination. At the trommel rotation speed of 47.37 r/min, the speed of the spike teeth reached 24.16 r/min, the inclination angle of the trommel was 5°, the impurity content was 6.0%, and the insect rate reached 1.2%. The results of the bench test indicated that the average impurity content was 5.87% and the average insect rate was 1.20%. The results of this study provide a reference for the improvement and optimization of the separation structure of the BSF sand mixture.

**Keywords:** black soldier fly; trommel screener; differential separation; design; test

## 1. Introduction

Livestock waste is accumulating with the expansion of the livestock breeding scale, which has resulted in serious pollution of the ecological environment. The use of insects to recycle livestock has been confirmed as a sustainable solution [1,2]. The application of BSF to dispose of livestock waste and to achieve resource utilization has become a focal point of current research [3–5]. The BSF sand mixture (the black soldier fly larvae (BSFL) and the BSF organic fertilizer) is harvested through the bioconversion of livestock waste, in which high-protein BSFL are employed as the protein feed of aquatic animals [6]. The BSF organic fertilizer sand is rich in active substances that act against plant diseases. It exhibits a high fertility and has served as an effective alternative fertilizer source for fruits, vegetables, and herbage [7]. At present, researchers worldwide have primarily focused on the bioconversion efficiency of BSF [8,9], the growth changes in BSF [10,11], the application of BSF organic fertilizers [12–14], the efficacy of BSFL protein [15,16], and BSFL that are

rich in amino acids [17,18]. Moreover, considerable research and exploration have been performed. The research results have further confirmed that the application of BSF to build a saprophytic chain is a vital measure facilitating the resource utilization of livestock waste.

Two-step and three-step earthworm separators were developed to examine the optimal harvest parameters and the key kinetic mechanism so as to shorten the harvest time of vermicompost fields and reduce the separation costs [19,20]. Compared with earthworms, the BSF sand mixture includes a certain moisture content, and some material particles are bonded to each other, which exhibit complex characteristics and significant differences in composition. The conventional screening technology and equipment have a low screening efficiency and are prone to blocking, thus limiting the resource utilization of the BSF sand mixture. Researchers worldwide have focused on the separation and cleaning of waste plastic film and impurities [21], cabbage seeds [22], soybean seeds [23], corn [24], rapeseed [25], and other materials over the past few years. To be specific, pneumatic screens, vibrating screens, trommel screeners, and other separation or sorting devices have been designed. The effects of the sieve size, type, arrangement, and operating parameters on the separation effect have been investigated, and a good screening efficiency and some theoretical research results have been achieved. Additionally, some research has considered factors (e.g., the wet agglomerating morphology change [26], blocking failure [27,28], bionic screen [29–31], and air flow [32–34]) and explored their effects on screening. Nevertheless, the results of existing experiments have suggested that the physical properties of the respective components are complex, and the morphology is different due to the adhesion of some particles in the BSF sand mixture [35] and the soft tissue of the BSFL [36]. The mechanical-physical properties and aerodynamic properties of the sieving objects are significantly different from those of the conventional screening objects. Accordingly, the separation effect of the existing screening equipment remains poor. The BSFL are easily damaged by external forces (e.g., extrusion and impact). Moreover, the BSFL are mixed with sand, which increases the difficulties in screening, thus resulting in the increased impurity rate in the BSFL. Biological separation is not satisfactory compared with mechanical separation. The BSFL can be lured or stressed through external environmental regulation. Furthermore, BSFL are capable of actively crawling and being separated from the base material, including bait trapping [37], light separation [38], thermal separation [39], water separation [39], electric field separation [40–42], etc. However, the separation efficiency completely depends on the wriggling and crawling of BSFL. Such conventional harvesting methods are time-consuming, slow, high in cost, and inefficient, limitations which do not apply to the large-scale production and application of the black soldier fly.

In accordance with existing research, focusing on BSF sand with a high water content formed through the biotransformation of pig manure by the BSF, this study developed a tubular sieve surface separation differential trommel screener to address the low screening efficiency and reduce the effect of sifting. Single-factor tests and multi-factor orthogonal tests were performed based on a differential trommel screener test bench to analyze the factors affecting the impurity content and the insect rate, so as to provide technical support and theoretical references for the screening of BSF sand mixture.

## 2. Materials and Methods

### 2.1. Black Soldier Fly Sand and Its Adhesion Nodule Model

The BSF sand mixture that was separated in this study was a product formed by the biotransformation of pig manure by black soldier flies, of which the main components were black soldier fly sand, BSFL. To be specific, the black soldier fly sand with the highest content was followed by the BSFL, with a mass ratio of 8 to 10 (Figure 1). The particle size distribution and percentage of the content of BSF sand were obtained using the sieving method. Most of the sand particles of the BSF were similar spheres, and a few were in the form of agglomerates. The particle size distribution was <1.6 mm, 1.6–2.6 mm, 2.6–3.5 mm, and >3.5 mm, respectively. The mass and mass fraction were 148.24 g, 18.53%, 383.68 g, 47.96%, 154.86 g, 19.36%, 113.22 g, and 14.15%, respectively. At the early stage, 50 BSFL

larvae were randomly examined. The cross-section of the BSFL was approximately oval. The BSFL physical parameters achieved the following mean values: a body length of 23.30 mm, standard deviation of 0.70 mm; body width of 4.90 mm, standard deviation of 0.34 mm; and the mass of 0.177 g/piece.

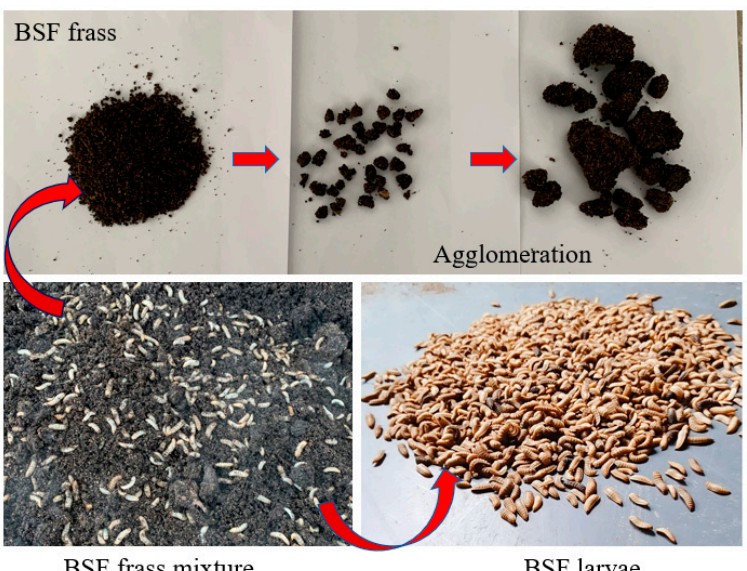

**Figure 1.** The main components of black soldier fly feces.

A certain amount of water exists on the sand particle surface of BSF. The adhesion between the sand particles of the BSF increases due to the action of liquid bridge force. Accordingly, the particles are easily bonded to each other to form a ball, thus leading to the increased particle size of the BSF sand. The round-hole screen and the grid-shaped screen on the conventional trommel are prone to this phenomenon, thus reducing the screen penetration rate of the trommel screener. There is adhesion between the sand particles of the BSF due to its moisture content. The particles form secondary and tertiary agglomerated particles during the movement process, thus resulting in the increased particle size. The adhesion process of BSF sand particles is illustrated in Figure 2. Figure 3 presents the established BSF sand particle adhesion nodule model.

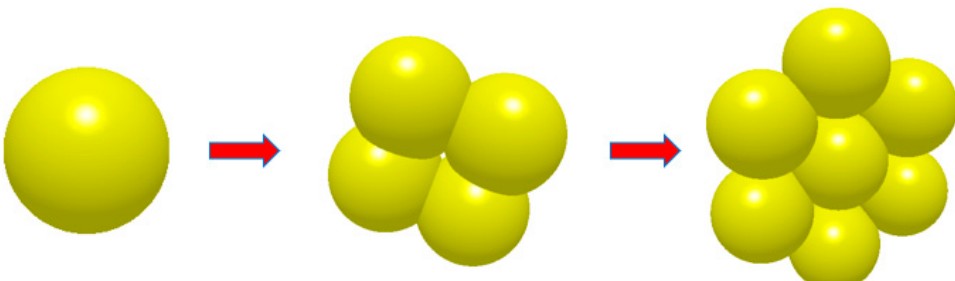

**Figure 2.** Process of black soldier fly feces adhered to become agglomerations.

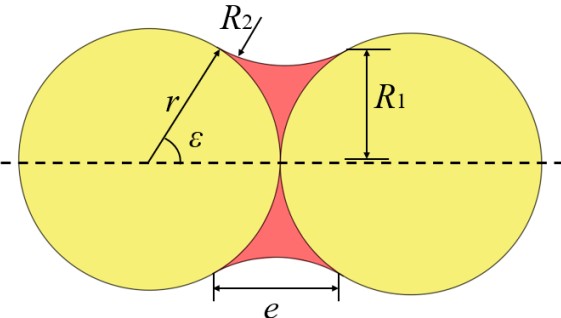

**Figure 3.** A model of black soldier fly feces binding into spheres.

Assuming that the contact angle between particles is zero, the adhesion force *F* between the sand particles of the BSF can be calculated as follows:

$$F = \pi R_1^2 \tau \left( \frac{1}{R_2} - \frac{1}{R_1} \right) + 2\pi R_1 \tau = \pi R_2 \tau \left( \frac{R_1 + R_2}{R_2} \right) \tag{1}$$

$$R_1 = r(\sec \varepsilon - 1) \tag{2}$$

$$R_2 = r(\tan \varepsilon - \sec \varepsilon + 1) \tag{3}$$

Based on Equations (1)–(3), this yields:

$$F = \frac{2\pi r \tau}{1 + \tan \frac{\varepsilon}{2}} \tag{4}$$

where $\tau$ denotes the surface tension; $r$ is the radius of the BSF sand particle; and $\varepsilon$ represents the pincer angle.

The analysis of Equation (4) indicates that when the BSF sand particles are in adhesive contact, the adhesion force between the particles is correlated with several factors (e.g., the radius of the particles, the water content, and the size of the pincer angle). For $N$ particles in a space with a certain volume of the trommel screener $V$, the probability $P$ that any two particles will collide with each other in time $t$ is calculated as follows:

$$P = \frac{4\pi r^2 t v_r N}{V} \tag{5}$$

where $N/V$ is the approximate particle concentration of the BSF sand material, and $v_r$ denotes the relative movement speed of the particles.

### 2.2. Structure and Working Principle
### 2.2.1. Overall Structure and Parameters

Figure 4 presents the structure of the designed ground trough BSF sand conveying and sorting machine. This machine was equipped with a wheeled, electric, self-propelled chassis, and it comprised a side-by-side double-impeller collecting device, a conveying device, and a sorting device. The organic fertilizer in the layered state, formed by the livestock manure that is bio-transformed by the BSFL and cultivated in the ground trough, can be aggregated, lifted, transported, sorted, classified, and utilized. The side-by-side double-impeller collecting device, i.e., the recovery structure of the whole machine, primarily achieves the orderly shoveling, transfer, and transportation of the insect sand in the layered state. The main technical parameters of the BSF sand aggregating, conveying, and sorting mechanisms were obtained through the preliminary test [43]. To be specific, the power of the impeller drive motor reached 120 w, the power of the driving motor of the walking chassis was 250 w, and the mass of the whole machine reached 55 kg (with the sorting device excluded). The aggregating width was 0.80 m, and the aggregating depth reached 0.15 m. The wheels were solid tires, the wheel diameter was 0.15 m, the wheelbase

was 0.75 m, and the overall dimensions of the machine (length × width × height) reached 2.50 m × 0.80 m × 0.80 m (the side-by-side double impeller extended to the limit position).

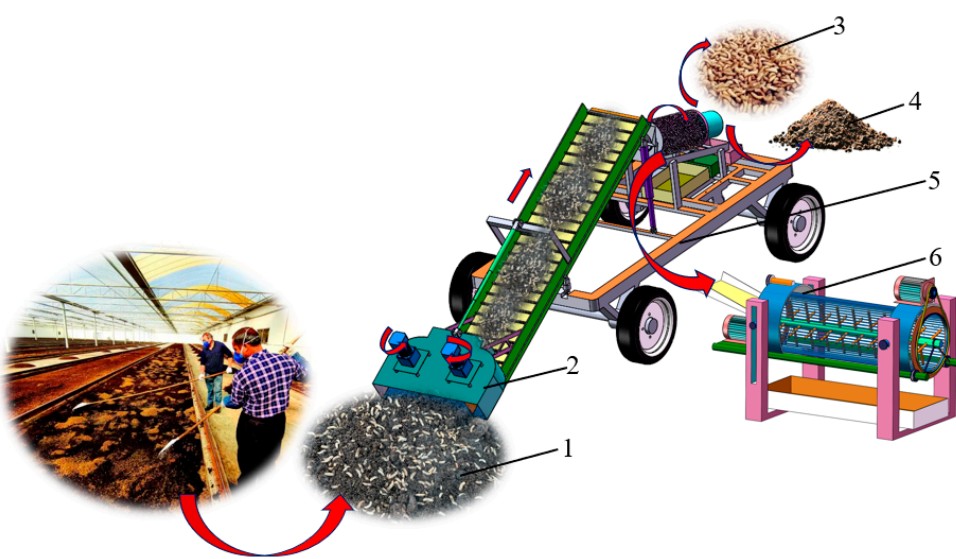

**Figure 4.** Structure and working principle of the ground trough BSF sand conveying and sorting machine. 1—BSF mixture; 2—parallel double-impeller collecting device; 3—BSF larvae; 4—BSF frass; 5—chassis; 6—sieve device with different rotational speeds.

### 2.2.2. Key Structure and Parameters of the Differential Trommel Screener

Figure 5 presents the overall structure of the differential trommel screener, which was the key structure for the separation, classification, and utilization of the BSF sand mixture. It primarily comprised of the base, the speed regulating motor, the screen, the spike teeth, the collecting box, and other components. The screening process of the differential trommel screener was captured by a high-speed camera, and the screening dynamic image was recorded in real time by its connection to a computer.

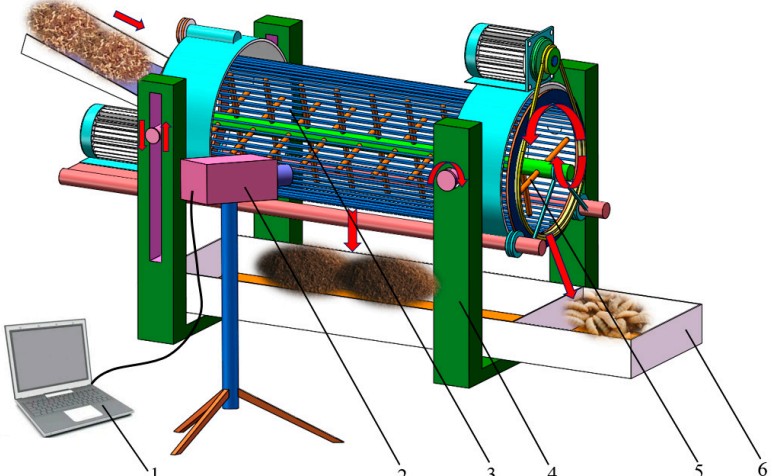

**Figure 5.** Cylinder sieve with promoting device with different rotational speeds. 1—computer; 2—high-speed camera; 3—tubular screen surface; 4—base; 5—spike teeth; 6—collection box.

The BSF sand easily agglomerates into clusters if the inner diameter of the trommel screener is too small. Moreover, a smaller inner diameter of the trommel screener will narrow the screening area of the trommel screener, whereas an inner diameter of the trommel screener that is too large will increase the power consumption of the screening operation. Given the installation space, the radius $R$ of the trommel screener was set

as 250 mm. In the internal screening process of the trommel screener, the design of the coaxial counter-rotating spikes can be beneficial for loosening, breaking, and stratifying BSF sand with a moisture content ranging from 40% to 50%, reduce the phenomenon of agglomeration and agglomeration, and increase the screening efficiency. The edges and corners of the general trapezoidal spike teeth may collide with the BSFL in the BSF sand mixture, thus causing damage to the surface of the insects. Accordingly, the spherical contact between the round-headed spike teeth and the BSF sand was established in order to reduce the rigid impact force on the BSF sand, increase the friction between the ends of the spike teeth and the BSF sand particles, and decrease the damage to the bodies of the BSFL caused by the spike teeth. The round-headed nail teeth were established as ball heads. Figure 6 presents the collision between the round-headed nail teeth and the BSFL.

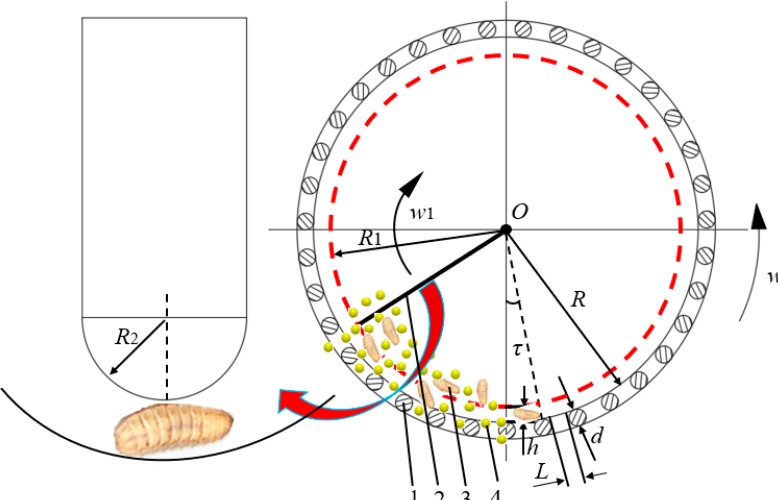

**Figure 6.** Schematic diagram of the cylinder sieve and nail tooth structure size. 1—tube screen; 2—nail tooth; 3—BSF larvae; 4—BSF sand particle.

Given the BSFL body and the particle size parameters of BSF sand, the gap between the spike teeth and the tubular screen surface was set to be slightly larger than the width of the BSL body to ensure that the BSF sand was fully in contact with the trommel screener, and to ensure that the BSFL would not be squeezed out through the gap of the tubular screen. In other word, $h$ was formulated to be 6 mm, the radius $R_1$ of the spike teeth around the rotation center was 244 mm, and the radius $R_2$ of the spike teeth ball head was 3 mm, which is smaller than the particle size of the agglomerated BSF sand particles, increasing the probability of the breaking of the sticky sand particles. The outer diameter of the tubular screener was smaller than the diameter of the agglomerated particles, and a solid stainless steel round tube with a diameter of 3 mm was established to reduce the possibility of larger BSF sand particles adhering to the surface of the tubular screener. To prevent the BSFL from leaking out of the gap on the surface of the tubular screener, the gap was large enough to allow the BSF sand particles to pass through the screener. The gap $L$ between the two adjacent circular tubes was set to 4 mm, and the angle between the two adjacent circular tubes was $\tau$, which is expressed as follows:

$$L = \frac{\pi R \tau}{180°} - d \tag{6}$$

where $\tau$ was calculated as 1.605°, and the angle $\tau$ between the two adjacent tubes was set as 1.5° for the sake of convenient machining and installation.

Existing research [44,45] has suggested that the design feed rate per unit length of the axial flow trommel screener is 2 kg/(s m). Given the wet characteristics of BSF sand, the design feed per unit length of the BSF sand trommel screener was 1.5~2 kg/(s m), and the length of the trommel screener was set as 1~1.3 m. In view of the installation space and the

structural size, the longer the trommel is, the greater the power consumption will be. The length of the trommel screener was set to 1.0 m.

### 2.2.3. The Principle of Separation

The side-by-side aggregating device moved forward at a constant speed along the inner side of the ground trough, and the inclination of the side-by-side double-impeller aggregating device and the horizontal plane was adjusted by the electric push rod. Moreover, the impeller speed of the side-by-side aggregating device and the forward speed of the whole machine were adjusted to complete the shoveling, lifting, and transfer along the ground trough, and a continuous material flow was provided for the differential trommel screener. In the differential trommel screener, the screen mesh and the spike teeth rotated differentially, and their rotation directions were opposite to one another. The trommel screener had a certain inclination in the axial direction, and the material in the trommel screener sloped down to the end discharge port. The screen was set as a circular array with equal spacing. The surface of the circular tube was a smooth arc surface without edges and corners, thus reducing the impact strength and impact force on the BSFL. The installation direction of the circular tube was consistent with the axial direction. The tubular screen ensured that the BSF sand particles and sundries were sifted through the gap of the circular tube while solving the problems of conventional tubular sieves, square mesh sieves, and smearing sieves. Furthermore, the spike teeth rotating coaxially and counter-rotating with the tubular screener facilitated the breaking, stratification, flow, and separation of the BSF sand mixture in the form of differential motion during the rotation process, thus preventing the BSF sand from sticking to create lumps and smearing the sieves, increasing the sieving efficiency.

### 2.3. Key Component Parameter Design and Analysis

### 2.3.1. The Effect of the Rotation Speed on the Material Sieving

The BSF sand mixture particles rise to a certain height on the trommel screen surface under the combined actions of the friction force of the screen surface and its own centrifugal force, and the particles finally roll down to the bottom of the screen surface under the action of gravity in a cyclic, periodic motion. Under the determined diameter, length, screen size and form, material, and other parameters of the trommel screener, the penetration rate of the mixed material in the trommel screener is correlated with the rotation speed of the trommel screener. The movement force in the trommel screen of the BSF sand particles was analyzed in this study (Figure 7) to ensure that the mixed material particles could pass through the effective screening area in time and reduce the screening loss.

At the highest point, the force of the BSF sand particles on the trommel screener surface is illustrated in Figure 7(I):

$$\begin{cases} mg \cos \beta + N_2 - \frac{mw^2 D}{2} = 0 \\ f_2 - mg \sin \beta = 0 \\ f_2 = \mu N_2 \end{cases} \tag{7}$$

Through calculation, this yields:

$$w = \sqrt{\frac{2g(\mu \cos \beta + \sin \beta)}{\mu D}} \tag{8}$$

To prevent the BSF sand particles from being discharged from the top of the trommel screener, $\beta$ should be greater than 0, i.e., when $\beta = 0$, the BSF sand particles will be discharged from the top in the vertical direction. Accordingly, when the trommel screener rotates, it is necessary to ensure that the gravity of the BSF sand particles is higher than

the centrifugal force, such that the BSF sand particles can fall back to the bottom, which is expressed as follows:

$$mg - \frac{mw^2 D}{2} > 0 \tag{9}$$

Through calculation, this yields:

$$w < \sqrt{\frac{2g}{D}} \tag{10}$$

Moreover, to ensure that the BSF sand particles can pass through the sieve holes of the trammel screen, the force of the particles in the sieve holes should meet the requirements (Figure 7(II)):

$$\begin{cases} \frac{mw^2 D}{2} + mg\cos\theta - f_1 > 0 \\ N_1 - mg\sin\theta = 0 \\ f_1 = \mu N_3 \end{cases} \tag{11}$$

Through calculation, we obtain:

$$w > \sqrt{\frac{2g(\mu\sin\theta - \cos\theta)}{D}} \tag{12}$$

Equations (9)–(12) indicate that the highest point of the trommel screener is affected by the rolling friction coefficient $\mu$, inner diameter of the trommel screener $D$, the angle $\beta$ between the line connecting the BSF sand particles and the center of the circle and the vertical center line, etc.

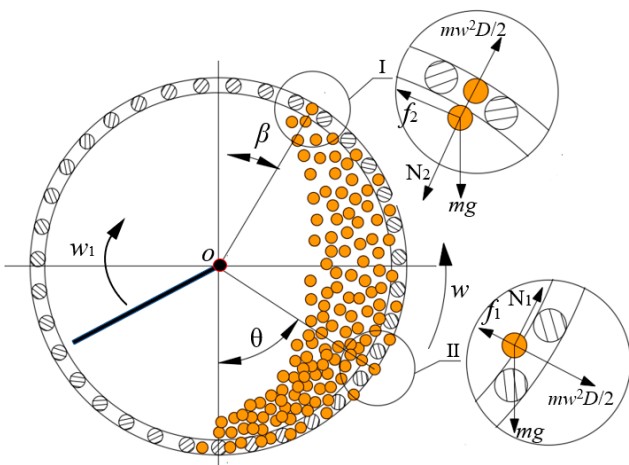

**Figure 7.** Stress analysis of BSF sand at the critical location. Note: $m$ denotes the BSF sand mass, g; $w$ is the angular velocity of cylinder sieve, rad·s$^{-1}$; I and II represent the low and high points, respectively; $w_1$ represents the angular velocity of the spike teeth, rad·s$^{-1}$; $\beta$ is the angle between the lines connecting the BSF sand at the highest point and the circle center and the vertical center line in a force balance state, rad; $\theta$ expresses the angle between the lines connecting the BSF sand passing through the sieve holes and the circle center with the vertical center line, $\theta \in (\alpha, \pi - \beta)$, rad; $D$ is the internal diameter of the cylinder sieve, m; $f_2$ is the friction at the highest point, N; $N_2$ is the supporting force at the highest point, N; $f_3$ is the friction when the rapeseed passes through the sieve holes, N; and $N_3$ is the supporting force when the rapeseed passes through the sieve holes, N.

The angle $\beta$ between the line connecting the BSF sand particles and the center of the circle and the vertical center line are correlated with the rotational angular velocity $w$ of the screener under the determined friction coefficient $\mu$ between the sand material and the screener surface, as well as the obtained inner diameter of the roller screen $D$. Thus, the

rotation speed of the trommel screener can affect the penetration position of the BSF sand particles on the screener surface and the screening efficiency.

Given the installation space and actual use of the whole machine, the inner diameter of the trommel screener is $D = 500$ mm. In reference to this [35], the friction coefficient $\mu$ between the BSF sand and the screen surface is selected as 0.4, $0 < \theta < \pi$. The rotation speed range of the trommel screener is expressed by substituting $\mu$ into Equations (9)–(12).

$$37.8 \text{ r/min} < n < 59.8 \text{ r/min}$$

### 2.3.2. Effect of the Spike Teeth Parameters on Material Impacting through the Screener

Four rows of spike teeth were designed on the rotating axis of the differential trommel screen, and the spike teeth were installed in a staggered manner at 90° to prevent the BSF sand materials from being affected by the axial side-by-side obstruction of the adjacent spike teeth (Figure 8).

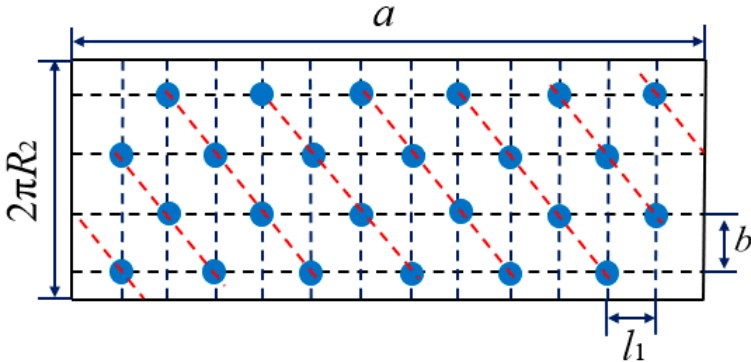

**Figure 8.** Arrangement of the nail teeth. Note: $a$ denotes the length of the rotating axis of the trommel screener, m; $l_1$ represents the teeth trace distance, m; $R_2$ is the radius of the rotation axis, m; and $b$ expresses the spike teeth spacing, m.

Above, $a$, with a value of 1.0 m, expresses the length of the rotating axis of the trommel screener. Given the body parameters of the BSFL, the teeth trace distance $l_1$ is 15 mm, which is less than the average body length of the BSFL, which is beneficial for loosening the BSFL and sand materials. The radius $R_2$ of the rotation axis, which was a solid round tube in our experiments, was 12.5 mm. The number of spike teeth increased, and the frequency with which the spike teeth impacted the BSFL increased with the excessively small spike teeth spacing $b$, thus reducing the contact time with the screener surface. Since the separation object was the BSF sand, a staggered design of 90° was selected for the spike teeth, i.e., $b$ was 19.625 mm to reduce the repeated impact on the material in the radial direction of the screener and to increase the drop flow frequency.

Simplifying the collision between the spike teeth and the BSF sand particles, the analytical model of the collision between the BSF sand particles and the spike teeth is illustrated in Figure 9.

As shown in Figure 9, a coordinate system was built with the rotation center O of the trommel screener as the origin. The horizontal direction was the $x$-axis, and the vertical direction was the $y$-axis. OB was simplified as the spike teeth of the trommel screener. The spike teeth rotated clockwise around the point O at an angular velocity $w_1$. The trommel screener rotated counterclockwise at an angular velocity $w$. The circle $O_1$ was the simplified BSF sand particles. After the agglutinate BSF sand particles collided with the spikes, they moved obliquely at the speed $v$. The spikes were fixed on the built-in rotating shaft of the trommel screener. Driven by the motor, the motion state of the spike teeth remained unchanged after the collision. Above, OC is the position of the spike teeth when the spike teeth collide twice with the BSF sand, while $(x_o, y_o)$ denote the coordinates at the collision

point A. After the collision, the agglutinate BSF sand particles move obliquely at the speed $v$. At this time, the motion trajectory equation of the BSF sand particles is expressed as follows:

$$\begin{cases} x = x_o - v \sin \gamma t \\ y = y_o + v \cos \gamma - \frac{1}{2} g t^2 \end{cases} \tag{13}$$

where $t$ is time, $s$.

The function equation of the angle between the line connecting the BSF sand particles and the center of rotation and the horizontal $x$-axis is written as follows:

$$\varphi_{(t)} = \tan^{-1} \frac{y_o + v_1 \cos \gamma - \frac{1}{2} g t^2}{x_o - v \sin \gamma t} \tag{14}$$

The angle equation between the spike teeth and the horizontal $x$-axis direction is written as follows:

$$\varphi'_t = \varphi_{(t=0)} - wt \tag{15}$$

When $t = 0$, i.e., $\varphi_{(t)} = \varphi'_t$, we can obtain the position at which the spike teeth of the trommel screener and the BSF sand particles collided for the first time. The BSF sand particles moved in the direction perpendicular to the spike teeth when the BSF sand particles collided with the spike teeth. The speed was slightly greater than the linear velocity in the circumferential direction of the spike teeth. In this case, the BSF sand particles and the spike teeth began to separate, and the sand particles of the BSF adopted an oblique projectile motion. The spiked teeth were driven by the motor to rotate in a circular motion at a uniform speed in the screener. After a certain period, they caught up with the sand particles of the BSF that were briefly separated. Subsequently, a secondary collision occurred. The time $t$, at this time, represents the second solution of $\varphi_{(t)} = \varphi'_t$ until the time when the BSF sand particles moved out of the collision area where they came into contact with the spike teeth, and the sieving was completed.

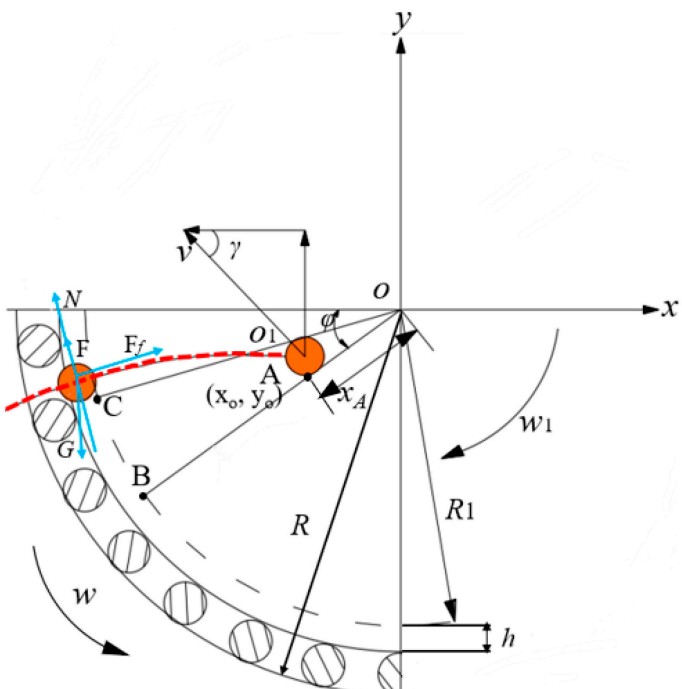

**Figure 9.** Collision force analysis between the nail teeth and BSF sand. Note: $G$ denotes the self-gravity of the BSFL, N; $F_f$ is the frictional force of the BSFL at the end of the spike teeth, N; F represents the effect of the spike teeth on the BSFL when the spike teeth rotated with a high-speed force, N; N is the support force of the spiked teeth on the BSFL, N; $w$ is the angular velocity of the cylinder sieve, rad·s$^{-1}$; and $w_1$ is the angular velocity of the spike teeth, rad·s$^{-1}$.

The analysis of Figure 9 suggests that the force *F* on the BSFL when the spike teeth rotate at high speeds is expressed as follows:

$$F = mw_1^2 R_1 \tag{16}$$

The force of the spikes on the BSFL must be less than 2.94 N [43], which is the limit of the force that the BSFL are able to withstand, measured by the physical property analyzer. In accordance with $w_1 = 2\pi n$, the rotational speed *n* of the spike teeth must fulfill:

$$n \leq 41.55 \, \text{r/min}$$

The analysis of the movement trajectory of the spike teeth and the BSF sand particles after the collision indicated that the spike teeth may undergo multiple collisions with the BSF sand particles. The rotational speed of the spike teeth affected the time of the collisions between the BSF sand particles and the spike teeth, thus further affecting the separation and sieving of the BSFL sand particles.

2.3.3. Effect of the Inclination Angle of the Trommel Screener on the Movement Trajectory of the Material

Figure 10 presents the movement trajectory of the BSF sand material. Its movement was divided into the linear motion in the direction of the rotary axis of the trommel screener and the plane motion perpendicular to the rotary axis of the trommel screener. These two types of motions were affected by the installation angle and the rotation speed of the trommel, respectively. The angle between the rotary axis of the trommel and the horizontal plane was $\sigma$ to ensure that the BSF sand material in the trommel had a certain fluidity in the axis direction and increase the probability of screen penetration. With a single particle *P* of the BSF sand placed at the origin O, a coordinate system was established (Figure 10). When the unit particle *P* of the BSF sand material entered the trommel from $D_1$, the particle immediately adopted a uniform circular motion, i.e., the trajectory line of the $D_1O$. The material was lifted to the O point, following the screener surface. The position of the O point was affected by certain parameters (e.g., the static friction coefficient of the BSF sand material, the rotation speed of the trommel, the inclination angle of the trommel, and the inner diameter of the trommel). Next, the BSF sand particles began to leave the screener surface to adopt a parabolic motion, i.e., the trajectory line of the $OD_2$, and reached the highest point E and then fell back to the point $D_2$. In such a cycle, the material was discharged from the trommel screener surface under a certain inclination.

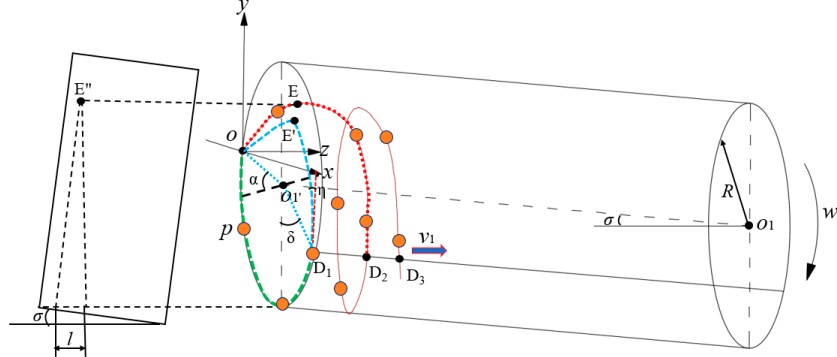

**Figure 10.** Kinematic analytical sketch of the BSF sand in the rotary screen. Note: $w$ denotes the angular velocity of the cylinder sieve, rad·s$^{-1}$; $w_1$ is the angular velocity of the spike teeth, rad·s$^{-1}$; $\sigma$ is the inclination angle of the trommel screen, °; $\eta$ expresses the angle between the trajectory of the landing point and the vertical direction, °; $R$ represents the radius of the cylinder sieve, m; $\alpha$ is the angle between the particle and horizontal direction, °.

The installation of the trommel screener was designed with a certain inclination angle. When the BSF sand particles in the trommel screener moved continuously, the motion

trajectory was approximately in the form of a spiral. The two adjacent drop points on the screener surface, at the bottom, were not in a fixed position, and they were separated by a certain distance along the axis of the trommel, and the distance between the two adjacent drop points was correlated with the inner diameter of the trommel, the rotation speed, and the inclination of the trommel. To be specific, the trajectory line $OE'D_1$ was the plane projection of the trajectory $OED_2$ in the plane perpendicular to the rotation axis of the trommel. Meanwhile, η expressed the angle between the trajectory of the landing point and the vertical direction. It is assumed that the BSF sand particles do not slide axially during the movement of the trommel. The moving distance ($D_1D_2$) of the two adjacent landing points of the BSF sand particles in the trommel in the axial direction of the trommel screen is expressed as follows:

$$l = 2R \tan \sigma \qquad (17)$$

where $l$ denotes the moving distance of the two adjacent landing points of the particle in the axial direction of the trommel, m.

In accordance with the existing analysis, the BSF sand particles can undergo two main stages within one motion cycle, which are the circular motion and parabolic motion, respectively. The circular motion time $t_1$ and the parabolic motion time $t_2$ are written as follows:

$$t = t_1 + t_2 = \frac{1}{w} \left( \sin \alpha \cos \alpha + [(\sin \alpha \cos \alpha)^2 + 2 \sin \alpha \sqrt{\sin \alpha + \cos \delta}] \right) \qquad (18)$$

Thus, the velocity of the particles of the BSF sand moving forward along the axis of the trommel is expressed as follows:

$$v_1 = \frac{l}{t} \qquad (19)$$

The analysis of the formula and Figure 10 indicated that the moving distance per unit time of the sand particles $l$ in the trommel screen was affected by the material movement and the parabolic motion process at the circular motion stage. Moreover, $l$ was directly correlated with the inclination angle $\sigma$ of the trommel screen, the rotation speed $w$ of the trommel screen, etc. In accordance with the existing trommel screen data available worldwide, combined with the previous pre-test data [46], the installation inclination angle of the trommel screen of 5~15° is more suitable.

### 2.4. Test Conditions

Test material: 4-day-old BSFL were placed in fresh pig manure with a moisture content of 70%. A BSF sand mixture was formed after 8 days of transabdominal biotransformation. The water content of the BSF sand was 44.36% (excluding the BSFL). The mass ratio of the material to insects in the BSF sand was set as 8. The separation test of the BSF was performed at the comprehensive training center of Hunan Agricultural University (Figure 11).

Test equipment: A Shengbang horizontal speed regulating motor (6IK250RGN-CF), electronic scale (YB502 type, accuracy of 0.01 g, Shanghai Haikang Electronic Instrument Factory, Shanghai, China), digital display revolution meter (DT-2236 type, accuracy of ±0.05%), three-measure digital display inclinometer (measurement range: 0~900°, accuracy of ±0.2°), stopwatch, digital video camera, etc., were applied.

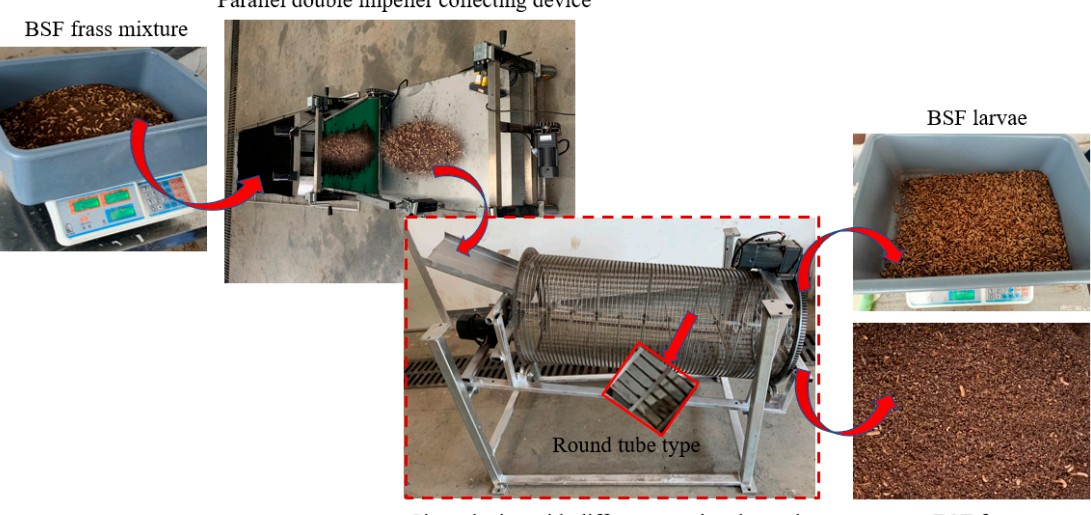

**Figure 11.** Test platform for the BSF frass mixture.

*2.5. Test Methods and Indicators*

The rotation speed of the trommel screen, the rotation speed of the spike teeth, and the inclination angle of the trommel screen were adjusted to the required values before the test. Subsequently, a certain quantity of the BSF sand mixture was placed in the feeding port of the trommel screen through the hopper to complete the feeding, separation, and other processes of the BSF sand material. At the end of the experiment, the mass of the organic manure of the BSF sand (excluding the BSFL) under the screener and the mass of the BSFL separated from the end of the screener were collected, respectively. The experiment was repeated 3 times for each group, and the average value was taken. To be specific, the trommel screener speed and spike speed were adjusted by the Shengbang horizontal speed regulating motor and measured by the digital display revolution meter. The inclination angle of the trommel screener was adjusted by the telescopic bracket, and the specific angle value was measured by the three-measure digital display inclinometer.

The test factors for the BSF sand separation performance included the inclination angle of the trommel screen, the rotation speed of the trommel screen, and the rotation speed of the spike teeth. Taking the impurity content in the insects and the insect impurity rate as the evaluation indexes, the calculation formula of the evaluation indexes is as follows:

$$Y_1 = \frac{m_1}{m_2} \times 100\% \tag{20}$$

where $Y_1$ is the impurity content in the insects, %. $m_1$ represents the mass of the organic fertilizer in the material at the discharge port of the trommel screen, g. $m_2$ expresses the total mass of the sand mixture at the discharge port of the trommel screen, g. We can also observe that:

$$Y_2 = \frac{m_3}{m_4} \times 100\% \tag{21}$$

where $Y_2$ is the insect impurity rate, %. $m_3$ is the total mass of the material under the trommel screener, g. $m_4$ is the mass of the BSFL in the material under the trommel screener, g.

*2.6. Experimental Design*

According to Equations (20) and (21) and comprehensive measurements of the actual requirements of the BSF sand separation operation, the impurity content in the insects and the insect impurity rate are used as the response values. The rotation speed, the speed of the spike teeth, and the inclination of the trommel screen were examined and studied using the Box–Behnken experimental design principle. Design-Expert 8.0.6 software was used to carry out a three-factor and three-level combination test, and the combination optimization

of the main parameters affecting the impurity content and the insect impurity rate was performed. After establishing the spike teeth and the inclination angle of the trommel screener and the basic data obtained from the previous test, the trommel rotation speed was 38~58 r/min, the spike tooth rotation speed was 10~40 r/min, and the inclination angle of the trommel screener reached 5~15° based on the theoretical analysis of the trommel screener. The test factor codes are listed in Table 1.

**Table 1.** Test factor code.

| Code Value | Factors | | |
| --- | --- | --- | --- |
| | Drum Rotation Speed $x_1$/(r·min$^{-1}$) | Nail Tooth Speed $x_2$/(r·min$^{-1}$) | Roller Screen Inclination $x_3$/(°) |
| −1 | 38 | 10 | 5 |
| 0 | 48 | 25 | 10 |
| 1 | 58 | 40 | 15 |

## 3. Results

### 3.1. Model Establishment and Significance Test

The test results are shown in Table 2 ($X_1$, $X_2$, and $X_3$ are the factor code values), and the regression fitting analyses of the impurity content and the insect impurity rate were conducted using the Design-Expert 8.0.6 software. The regression equations of the impurity content $Y_1$ and the insect rate $Y_2$ were established to conduct the significance test and analysis of the three factors for the test indicators. Lastly, the quadratic polynomial response surface regression model of the significant test factors and evaluation indicators was obtained. The model significance test results are listed in Table 3.

**Table 2.** Test design and response results.

| Number | $X_1$ | $X_2$ | $X_3$ | $Y_1$/% | $Y_2$/% |
| --- | --- | --- | --- | --- | --- |
| 1 | −1 | −1 | 0 | 8.1 | 3.3 |
| 2 | 1 | −1 | 0 | 10.2 | 2.8 |
| 3 | −1 | 1 | 0 | 9.7 | 2.7 |
| 4 | 1 | 1 | 0 | 9.5 | 2.2 |
| 5 | −1 | 0 | −1 | 6.5 | 2.4 |
| 6 | 1 | 0 | −1 | 7.3 | 2.3 |
| 7 | −1 | 0 | 1 | 6.3 | 3.5 |
| 8 | 1 | 0 | 1 | 7.3 | 2.3 |
| 9 | 0 | −1 | −1 | 8.4 | 1.1 |
| 10 | 0 | 1 | −1 | 9.2 | 1.1 |
| 11 | 0 | −1 | 1 | 7.8 | 1.8 |
| 12 | 0 | 1 | 1 | 8.3 | 1.1 |
| 13 | 0 | 0 | 0 | 5.2 | 1.5 |
| 14 | 0 | 0 | 0 | 5.9 | 1.9 |
| 15 | 0 | 0 | 0 | 5.6 | 1.7 |
| 16 | 0 | 0 | 0 | 5.3 | 1.6 |
| 17 | 0 | 0 | 0 | 5.7 | 1.8 |

**Table 3.** Significance test of model.

| Source | \multicolumn{4}{c}{Impurity Content in BSF Larvae} | | | | \multicolumn{4}{c}{Insect Impurity Rate} | | | |
|---|---|---|---|---|---|---|---|---|
| | df | Mean Square | $F_1$ | $P_1$ | df | Mean Square | $F_2$ | $P_2$ |
| Model | 9 | 4.70 | 58.32 | <0.0001 | 9 | 0.92 | 39.42 | <0.0001 |
| $X_1$ | 1 | 1.71 | 21.22 | 0.0025 | 1 | 0.66 | 28.48 | 0.0011 |
| $X_2$ | 1 | 0.61 | 7.50 | 0.0290 | 1 | 0.45 | 19.44 | 0.0031 |
| $X_3$ | 1 | 0.36 | 4.48 | 0.0721 | 1 | 0.41 | 17.45 | 0.0042 |
| $X_1X_2$ | 1 | 1.32 | 16.40 | 0.0049 | 1 | 0.000 | 0.000 | 1.0000 |
| $X_1X_3$ | 1 | $1 \times 10^{-2}$ | 0.12 | 0.7351 | 1 | 0.30 | 13.03 | 0.0086 |
| $X_2X_3$ | 1 | 0.022 | 0.28 | 0.6137 | 1 | 0.12 | 5.28 | 0.0552 |
| $X_1^2$ | 1 | 5.38 | 66.67 | <0.0001 | 1 | 6.06 | 261.18 | <0.0001 |
| $X_2^2$ | 1 | 30.81 | 382.04 | <0.0001 | 1 | 0.095 | 4.08 | 0.0831 |
| $X_3^2$ | 1 | 0.14 | 1.69 | 0.2346 | 1 | 0.32 | 13.72 | 0.0076 |
| Residual | 7 | 0.081 | | | 7 | 0.023 | | |
| Lack of fit | 3 | 0.077 | 0.93 | 0.5025 | 3 | 0.021 | 0.83 | 0.5413 |
| Pure error | 4 | 0.083 | | | 4 | 0.025 | | |

In accordance with Table 3, the response surface models $P_1$ and $P_2$ of the impurity content and the insect rate are both less than 0.0001, indicating that the regression model is extremely significant. The lack of fit was 0.5025 and 0.5413, respectively, both being greater than 0.05. The lack of fit was not significant. In other words, the quadratic regression equation fitted by the model was highly consistent with the actual test, which reflects the relationship between the impurity content $Y_1$, the insect rate $Y_2$, and $X_1$, $X_2$, and $X_3$. The working parameters of the BSF sand can be effectively optimized by the regression model.

The regression equation of the impurity content is:

$$Y_1 = 5.54 + 0.46X_1 + 0.27X_2 - 0.21X_3 - 0.58X_1X_2 + 0.05X_1X_3 - 0.075X_2X_3 + 1.13X_1^2 + 2.71X_2^2 + 0.18X_3^2 \tag{22}$$

The regression equation of the insect rate is:

$$Y_2 = 1.70 - 0.29X_1 - 0.24X_2 + 0.23X_3 - 0.28X_1X_3 - 0.17X_2X_3 + 1.20X_1^2 - 0.15X_2^2 - 0.27X_3^2 \tag{23}$$

By analyzing the *p* value of each factor, the effect of each factor on the impurity content, in descending order, is: the rotation speed of the trommel screener, the speed of the spike teeth, and the inclination of the trommel screen. The effect of each factor on the insect rate, in descending order, is: the rotation speed of the trommel screener, the speed of the spike teeth, and the inclination of the trommel screen.

*3.2. Analysis of the Effects of Various Factors on the Indicators*

To analyze the effects of the rotation speed of the trommel screen, the rotation speed of the spike teeth, and the inclination angle of the trommel screen on the impurity content and the insect rate more intuitively, a response surface diagram of the relationship between the test index and each factor was created according to the quadratic regression model. The shape of the response surface reflects the effects of the interaction factors, as shown in Figure 12.

As depicted in Figure 12a, when the trommel rotation speed was fixed at a certain level, and the spike tooth rotation speed increased from 10 r/min to 40 r/min, the impurity content first decreased and then increased. The main reason for this phenomenon is that when the speed of the spike teeth is slow, part of the wet BSF sand has a low contact frequency with the spike teeth and is not sufficiently broken by the spike teeth. The ratio is relatively high, such that the impurity content is relatively high. When the speed of the spikes increased to 25 r/min, the sticky and wet agglomerated particles in the BSF sand mixture were effectively broken by the spikes, and the separation effect was the best. However, with the increase in the speed of the spike teeth, some of the sticky and wet BSF sand was impacted by the spike teeth, but it did not have enough time to slide down to

the trommel screener and be rolled up by the rotating spike teeth, resulting in the higher impurity content in the insects. When the rotation speed of the spike teeth was fixed, the impurity content in the insects first decreased and then increased sharply with the increase in the rotation speed of the trommel. The main reason for this is that the BSF sand mixture is constantly turned with the increase in the rotation speed of the trommel, and there is a large drop in the trommel screener, and the screening effect is better. At the trommel speed of 48 r/min, the screening effect was the best. With the increase in the rotation speed of the trommel screener, part of the BSF sand mixture easily attached to the wall of the trommel screen due to the centrifugal force. Furthermore, the internal relative movement was diminished, and most of the fine materials were difficult to screen out, resulting in the increase in the impurity content in the insects.

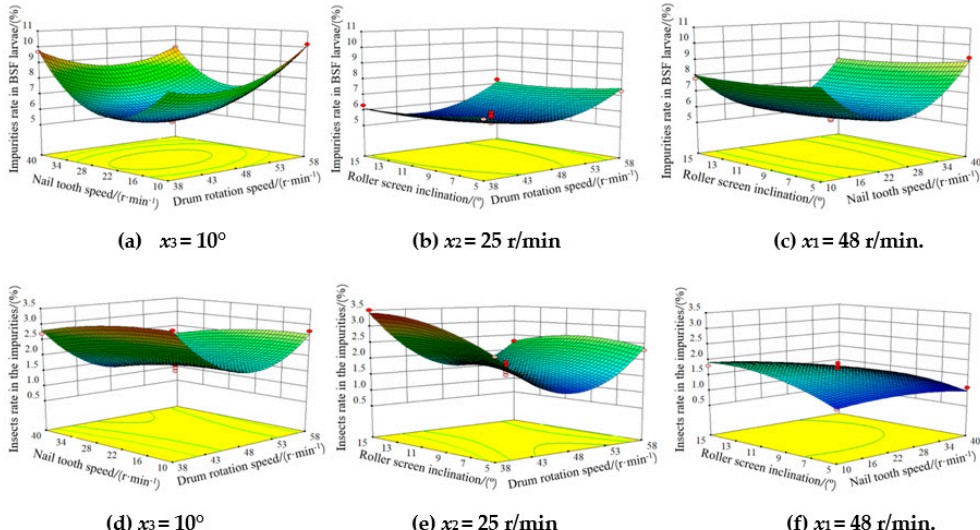

**Figure 12.** Effects of various factors on the membrane impurity separation performance.

The shape and contour density of the response surface, shown in Figure 12b, indicated that the interaction between the trommel screener speed and the trommel screener inclination angle did not significantly affect the impurity content. The effect of the trommel speed on the impurity content was slightly stronger than that of the trommel screen inclination angle, which is consistent with the results of the variance analysis.

The response surface shape variation trend, shown in Figure 12c, indicated that the interaction between the spike teeth speed and the trommel screener inclination angle did not significantly affect the impurity content. To be specific, the trommel screener inclination angle slightly affected the impurity content. With the increase in the spike speed, the effect of the spike speed on the impurity content decreased first and then increased, which is consistent with the results of the variance analysis.

As depicted in Figure 12d, the interaction between the trommel screener speed and spike speed had different impacts on the insect rate. The analysis of the response surface shape and contour density indicated that the effect of the trommel screener speed on the insect rate was significantly greater than that of the spike speed. At the constant spike speed, with the increase in the trommel screener speed, the insect rate decreased first and then increased. However, at the fixed trommel screener speed, the insect rate changed slightly with the increase in the spike speed. The main reason for the above phenomenon is that when the rotation speed of the trommel screener is lower than 48 r/min, the BSF sand mixture is fully turned over and broken up with the increase in the trommel screener speed. In this case, the separation effect was significant. The organic fertilizer particles in the BSF sand efficiently passed through the screener, and the insect rate was low. However, when the rotation speed of the trommel screener was higher than 48 r/min, some BSF fell

through the gap of the trommel round tube sieve due to the excessive centrifugal force, thus resulting in the increase in the insect rate.

The response surface shape, shown in Figure 12e, indicated that the interaction between the rotational speed of the trommel and the inclination angle of the trommel screener significantly affected the insect impurity rate. The insect rate tended to increase with the increase in the trommel screener angle. The main reason for this is that when the length and rotation speed of the screener are fixed, the inclination angle increases, thus leading to the growing inclination of the BSF sand material's layer surface in the advancing direction of the material. The number of motion cycles in the screener decreases, which reduces the dispersing degree and sieving efficiency of the BSF sand mixture, thus leading to the increased insect rate in the mixture. However, when the trommel screen angle was fixed at a certain level, the trommel speed increased, and the insect rate first decreased rapidly and then increased. The reason is that with the gradual increase in the trommel speed, the BSF sand mixture inside the trommel screen, affected by the reverse movement, can be more effectively separated. Subsequently, the BSFL dropped from the end of the trommel screener. Nevertheless, when the rotation speed of the trommel increased from 48 r/min to a higher speed, the overturning centrifugal force formed by the screener increased. As a result, some BSFL were thrown out from the gap of the circular screener due to inertia, thus resulting in the increase in the insect impurity rate. Moreover, the results of the response surface shape analysis, shown in Figure 12e, indicated that the effect of the rotation speed of the trommel screener on the insect impurity rate was greater than that of the inclination angle of the trommel screener, which is consistent with the results of the variance analysis.

The results of the shape analysis of the response surface, shown in Figure 12f, indicated that the interaction between the inclination angle of the trommel screener and the speed of the spike teeth did not significantly affect the insect impurity rate, which is consistent with the results of the variance analysis. The effects of the two factors on the insect impurity rate were relatively small, and the surface response changed slowly. At the fixed speed of the spike teeth, the number of turning movements of the BSF sand mixture in the screener was reduced due to the increase in the inclination angle of the trommel screener. The residence time of the material was shortened, and the probability of the material penetrating the screener decreased, thus resulting in an increase in the insect impurity rate. At the fixed inclination angle of the trommel screener, the insect impurity rate changed slightly with the increase in the rotation speed of the spike teeth. The reason for this is that the frequency of the BSF sand mixture rolled up by the spike teeth increases when the speed of the spike teeth is 25 r/min. There was insufficient time for the BSF sand mixture to fall through the gap of the tubular screener, thus resulting in the ineffective screening effect of the BSF sand mixture. The insect impurity rate did not change significantly.

### 3.3. Parameter Optimization and Experimental Verification

To meet the requirements of the separation operation of the impurity content in the insects and the insect impurity rate, the Design-Expert 8.0.6 software optimization module was employed so as to achieve the multi-objective optimization of the trommel speed, the spike speed, and the inclination angle of the trommel screener during the differential screening operation. The objective function constraint conditions are expressed as follows:

$$s.t. \begin{cases} minY_1(X_1, X_2, X_3) \\ minY_2(X_1, X_2, X_3) \\ \begin{cases} -1 \leq X_1 \leq 1 \\ -1 \leq X_2 \leq 1 \\ -1 \leq X_3 \leq 1 \end{cases} \end{cases} \tag{24}$$

The optimal parameter combination was obtained using the optimization solution, as follows: the rotation speed of the trommel was 47.37 r/min, the rotation speed of the spike teeth was 24.16 r/min, and the inclination angle of the trommel was 5°. Under the

above conditions, the impurity content in the insects was 6.0% and the insect impurity rate was 1.2%.

To verify the feasibility of the optimization results, a validation test was performed at the comprehensive training center of Hunan Agricultural University according to the optimal working parameters. To facilitate the adjustment of the parameters, the trommel screener speed was set at 47 r/min, the spike speed at 24 r/min, and the trommel screen inclination angle at 5°. The test was performed three times, and the results indicated that the average insect impurity content was 5.87%, and the average insect impurity rate in was 1.20%. The experimental results were close to the predicted values, which verified the accuracy of the model.

## 4. Discussion

A side-by-side double-impeller aggregate conveying device was designed at the early stage of the study to achieve the effective collection and transfer of BSF sand [35]. On this basis, a screening device should be designed in order to effectively solve the sorting problem of the BSF sand mixture and manage resource utilization. The existing screening test results suggested that [46] the BSF sand mixture had a certain moisture content. Without stratification, the stirring and beating device, the conventional trommel screener, and the round-hole mesh screener have a lower screening efficiency for BSF sand mixture, which easily forms a paste on the screen and reduces the screener penetration, resulting in a higher insect rate and impurity content. Furthermore, within a certain moisture content range, the adaptability of the trommel screener according to the separation of the BSF sand mixture could be improved further.

The trommel screener, as a rotary motion screener, has been extensively used in the sorting of grain and other agricultural materials, and little research has been conducted on the separation of BSF mixture. Combined with the mechanical and physical properties of the BSF sand mixture, the layered rotating spike teeth were added to the trommel screener to facilitate the dispersion and stratification of the material. Unlike the conventional trapezoidal spikes, the contact between the spherical spikes and the BSFL is spherical, thus increasing the impact cross-sectional area and reducing the rigid impact on the BSFL. Moreover, the spike teeth and the trommel screener rotate coaxially and reversely, which is more conducive to reducing the probability that the BSF sand mixture will adhere to the screener and to improving the screener penetration rate compared with the conventional trommel screener.

The conventional trommel screener is a round-hole screener or mesh screener, achieving a higher screening efficiency for agricultural bulk materials with a low moisture content. However, for a BSF sand mixture with a high moisture content, the BSFL and insect sand materials exhibit a high viscidity, and the BSFL skin has a certain elasticity, thus actively blocking the sieve holes and creating a sieve paste phenomenon. The tubular screener, with an elongated gap, smooth surface, and no edges or corners, was used to solve the problem of the entanglement and clogging of the BSFL sand mixture that arises from the small sieve holes. Furthermore, the BSF sand mixture was formed in a thin layer of BSF sand on the inner side of the tubular screener due to the action of the coaxial and reverse rotating spikes, which can be effectively separated. The penetration of the BSF sand mixture was enhanced, and the insect impurity rate under the screener was reduced.

This study created some innovative structural designs, but there are also some limitations. The details are as follows:

(1) The differential trommel screener designed to separate the mixture of BSF sand was applied under the test conditions, and the trial production test was performed. If it is essential to complete the large-scale separation of BSF sand mixture, the size parameters of the trommel screener should be optimized in accordance with the feeding amount so as to better meet the actual production requirements.

(2) The moisture content of the BSF sand mixture varied because of the differences in the biotransformation of the BSFL, and the range of the moisture content fluctuated.

The differential trommel screen designed in this study had an excellent screening effect under the test conditions. For a BSF sand mixture with a higher moisture content, some operating parameters should be further optimized to increase the screening penetration rate. Furthermore, when the feeding amount of BSF sand mixture fluctuates, the adaptability of the differential trommel screener should be studied in depth.

## 5. Conclusions

(1) In accordance with the requirements for the separation of BSF sand mixture, a type of differential separation trommel screener was developed using a method combining theory and experiments. The trommel screener and the spiked teeth rotated coaxially and reversely, thus increasing the probability of the layering and sieving of the BSF sand mixture. The round-headed spike teeth effectively reduced the rigid impact on the BSFL. The relevant experimental factors for the insect rate and the impurity content were determined through the analysis of the movement of the BSF sand mixture in the differential trommel screener.

(2) According to the Box–Behnken experimental design principle, the three-factor and three-level response surface analysis method was adopted to perform the separation performance test of the differential trommel screener in separating the BSF sand mixture. Through the analysis of the response surface, it was found that the factors affecting the impurity content in the insects and the rate of insect impurities were the same and in the descending order as follows: the trommel rotation speed, spike teeth rotation speed, and inclination angle of the trommel screener.

(3) The quadratic polynomial regression models of the impurity content in the insects, the rate of insect impurities, the rotational speed of the trommel, the rotational speed of the spike teeth, and the inclination of the trommel screener were built, respectively. The optimal operation parameters of the differential separation trommel screener were obtained through optimization and solutions. To be specific, the rotation speed of the trommel was 47.37 r/min, the rotation speed of the spike teeth was 24.16 r/min, and the inclination angle of the trommel was 5°. Under the above parameters, the impurity content in the insects was 6.0%, and the rate of insect impurities reached 1.2%. By revising the optimized parameters, under the combination of the trommel speed of 47 r/min, the spike speed of 24 r/min, and the inclination angle of 5°, the average insect rate and impurity content reached 5.87% and 1.20%, thus satisfying the actual production demands and increasing the BSF sand separation efficiency.

**Author Contributions:** Methodology, investigation, project administration, supervision, funding acquisition, writing—review and editing, C.P.; conceptualization, resources, writing—original draft, T.Z.; investigation, data curation, S.S. (Shisheng Song); investigation, project administration, supervision, S.S. (Songlin Sun); project administration, Y.Y.; visualization, supervision, D.X. All authors have read and agreed to the published version of the manuscript.

**Funding:** This research was supported by the Natural Science Foundation of Hunan Province, China (grant No.2020JJ5253).

**Institutional Review Board Statement:** Not applicable.

**Informed Consent Statement:** Not applicable.

**Data Availability Statement:** All data are presented in this article in the form of figures and tables.

**Conflicts of Interest:** The authors declare no conflict of interest.

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
