# Peer review of "Design and Experiment of Black Soldier Fly Frass Mixture Separation through a Cylinder Sieve with Different Rotation Speeds"

_applsci, doi:10.3390/app122010597_

Round 1

Reviewer 1 Report

The research topic is interesting, the results are also interesting.

However, there would be some things to improve, from my point of view bibliographic sources should be added, to be around 45-50 bibliographic sources.

The images look very good and are very suggestive, they probably still need to be arranged a bit (the images should be properly centered) and some images lack explanations (figure 7, 10, 12).

Otherwise, the work looks fine and the images help a lot.

Reviewer 2 Report

The paper is correctly written from a scientific point of view. The data and the results are clearly presented. The test methods used are well described.

I recommend publishing this paper if the authors take into consideration the following: 

-         - The equations should be formatted according to the guideline for authors.

-        - The authors could consider improving the conclusions section in order to better emphasize their original contribution to the paper.
